# Biologics and Small Molecule Inhibitors for Treating Hidradenitis Suppurativa: A Systematic Review and Meta-Analysis

**DOI:** 10.3390/biomedicines10061303

**Published:** 2022-06-02

**Authors:** Chun-Hsien Huang, I-Hsin Huang, Cheng-Chen Tai, Ching-Chi Chi

**Affiliations:** 1Department of Dermatology, Chang Gung Memorial Hospital, Linkou Main Branch, No. 5, Fuxing Street Guishan District, Taoyuan City 333423, Taiwan; gshuang0614@gmail.com (C.-H.H.); eugenia9797@gmail.com (I.-H.H.); 2Medical Library, Department of Medical Education, Chang Gung Memorial Hospital, Linkou Main Branch, No. 5, Fuxing Street Guishan District, Taoyuan City 333423, Taiwan; litbeartwo@gmail.com; 3College of Medicine, Chang Gung University, No. 259, Wenhua 1st Road, Guishan District, Taoyuan City 333323, Taiwan

**Keywords:** hidradenitis suppurativa, biologics, small molecule inhibitors, systematic review, meta-analysis, adalimumab, bimekizumab

## Abstract

Background: The treatment guidelines for hidradenitis suppurativa (HS) vary among different countries, and several biologics and small molecule inhibitors have been tested for treating moderate-to-severe HS over the past few years. However, treatment guidelines for HS vary among different countries. Methods: A systematic review and meta-analysis was performed to exam the efficacy and serious adverse events (SAEs) of biologics and small-molecule inhibitors in treating moderate-to-severe HS. Binary outcomes were presented as risk ratio (RR) with 95% confidence interval (CI). Results: We included 16 RCTs with a total of 2076 participants on nine biologics and three small-molecule inhibitors for treating moderate-to-severe HS, including adalimumab, anakinra, apremilast, avacopan, bimekizumab, CJM112, etanercept, guselkumab, IFX-1, INCB054707, infliximab, and MABp1. The meta-analysis revealed only adalimumab (RR 1.77, 95% CI, 1.44–2.17) and bimekizumab (RR 2.25, 95% CI, 1.03–4.92) achieved significant improvement on hidradenitis suppurativa clinical response (HiSCR), and adalimumab was superior to placebo in achieving dermatology life quality index (DLQI) 0/1 (RR 3.97; 95% CI, 1.70–9.28). No increase in SAEs was found for all included active treatments when compared with placebo. Conclusions: Adalimumab and bimekizumab are the only two biologics effective in achieving HiSCR with acceptable safety profile, whereas adalimumab is the only biologic effective in achieving DLQI 0/1.

## 1. Introduction

Hidradenitis suppurativa (HS), also known as acne inversa, is a chronic inflammatory dermatosis presenting with recurrent painful subcutaneous nodules, abscess, and draining sinuses with typical distribution and may progress to scarring and impaired quality of life [1]. HS has several complications, including anemia, infection, and squamous cell carcinoma, and has been associated with coronary artery disease, inflammatory bowel disease, spondyloarthritis, pyoderma gangrenosum, psoriasis, anxiety, and depression [2,3,4,5,6]. The prevalence of HS was estimated 0.4% [7], which varies among different geographic regions with a female preponderance in North America and Europe but a male preponderance in Asia [8,9].

The pathogenesis of HS is multifactorial and not fully understood yet [10,11]. Mutations of nicastrin and proline–serine–threonine phosphatase-interacting protein 1 genes, upregulation of proinflammatory cytokines (including tumor necrosis factor (TNF), interleukin (IL)-17, and IL-23), altered microbiome, and physiological and environmental factors, such as obesity and smoking, are the four key factors that contribute to the pathogenesis of HS [12]. Because of its multifactorial etiology with unpredictable response to therapy, the treatment guidelines for HS vary among different countries. In general, the first-line therapies for mild-to-moderate HS or widespread Hurley stage I/II HS are tetracyclines or combination of clindamycin and rifampicin [13,14]. Other systemic treatments varied across several guidelines or consensus, including metronidazole/moxifloxacin/rifampicin triple therapy, dapsone, ertapenem, acitretin, and isotretinoin, with colchicine also proposed in the management of HS [15,16,17,18,19]. Biologics have been recommended for moderate-to-severe HS when conventional systemic therapies fail. Adalimumab is recommended across all guidelines and consensus as first-line biologic for moderate-to-severe HS unresponsive to systemic antibiotics [15,16,17,18]. Recently, many biologics and small molecule inhibitors have been tested for treating moderate-to-severe HS, defined as Hurley stage II/III, hidradenitis suppurativa physician global assessment (HS-PGA) moderate/severe, or hidradenitis suppurativa severity index (HSSI) score 8–12/≥ 13 [13,20,21,22]. The objective of this study was to evaluate the efficacy and safety of biologics and small molecule inhibitors in treating moderate-to-severe HS.

## 2. Materials and Methods

We performed a systematic review and meta-analysis to examine the effects of biologics and small-molecule inhibitors in treating moderate-to-severe HS, and followed the Preferred Reporting Items for Systematic review and Meta-Analyses (PRISMA) guidelines [23]. We have registered the study protocol with PROSPERO (CRD42021279316). This study was exempted from ethical review by the Chang Gung Medical Foundation (202002102B1).

### 2.1. Literature Search and Study Selection

We searched relevant studies in the Cochrane Library, Medline, and Embase databases and trials registered in ClinicalTrials.gov from their respective inception through 19 May 2022 with the assistance of an information specialist (C.-C.T.). The search strategy is listed in Appendix A.

Studies that met the following criteria were included: (1) randomized controlled trials (RCTs), which tested the efficacy and safety of biologics or small-molecule inhibitors in treating moderate-to-severe HS; and (2) studies that reported at least one of the following outcomes of our interest: hidradenitis suppurativa clinical response (HiSCR), serious adverse events (SAE), Sartorius score (SS), modified Sartorius score (MSS), physician global assessment (PGA), hidradenitis suppurativa physician global assessment (HS-PGA), and dermatology life quality index (DLQI). Studies other than RCTs were excluded, including post hoc analysis and open-label extension studies. When there were multiple reports of the same RCT, we included the first report and extracted usable data from the others. Two authors (C.-H.H. and I.-H.H.) independently screened the search results through scanning titles and abstract, followed by reviewing the full text of potentially eligible studies. Discrepancies were resolved by discussion with the supervising author (C.-C.C.).

### 2.2. Data Extraction and Risk of Bias Evaluation

One author (C.-H.H.) extracted the following data from included RCTs: (1) name of authors; (2) clinical trial number and name; (3) study period; (4) countries in which the trial was performed; (5) study protocol; (6) profile of patients (i.e., sample size, sex, age, and disease severity); (7) treatment regimen; (8) outcome assessment; and (9) SAE. Another author (I.-H.H.) rechecked the extracted data. We chose HiSCR at weeks 12 to 16 into therapy as primary outcome because it was the most widely used assessment tool among clinical trials and was supported by good-quality validated data [24]. Our secondary outcomes included SAE and other efficacy outcomes, for example DLQI and MSS at weeks 12 to 16. If a trial did not report outcomes measured between weeks 12 to 16, we analyzed outcome data available at other time points but did not include them in the meta-analysis.

Two authors (C.-H.H. and I.-H.H.) independently appraised the risk of bias of included RCTs according to the Cochrane Collaboration’s tool [25]. Disagreements were resolved by discussing with the supervisor (C.-C.C.) until a consensus was achieved.

### 2.3. Statistical Analysis

We used Review Manager version 5.4.1 (The Cochrane Collaboration, London, UK, 2020) to perform quantitative synthesis. Binary outcomes were presented as risk ratio (RR) with 95% confidence interval (CI). Continuous outcomes were presented as mean difference (MD) with 95% CI. The I^2^ was used for quantifying statistical heterogeneity. An I^2^ statistic of ≥ 50% was considered moderate-to-high statistical heterogeneity [26]. The random-effects model was adopted in conducting meta-analyses because of anticipated clinical heterogeneity. A *p* value of <0.05 was defined as significant. The meta-analysis on adalimumab only included data from the approved regimen, i.e., 160 mg at week 0, 80 mg at week 2, followed by 40 mg weekly or 80 mg every other week afterward [27]. We were unable to perform a subgroup or sensitivity analysis because most of the included biologics and small-molecule inhibitors were only tested in a single RCT. Publication bias was also not evaluated because the included studies measured in primary or secondary outcomes were <10 studies.

## 3. Results

The PRISMA study flow chart is presented in Figure 1. Our search identified 772 records from databases and 1 record registered in ClinicalTrials.gov, with 563 screened for the titles and abstracts after removing 210 duplicates. The full text of 40 reports were assessed for eligibility. After removing 5 open-label extension studies, 8 post hoc analysis studies, and 2 studies that assessed efficacy by measuring change of cytokines in lesional skin, we included 15 reports addressing 16 RCTs with a total of 2076 participants. Among the 16 RCTs, the efficacy and safety of nine biologics and three small-molecule inhibitors in treating moderate-to-severe HS were evaluated, including adalimumab (*n* of RCTs = 6) [21,28,29,30,31], anakinra (*n* = 1) [32], apremilast (*n* = 1) [33], avacopan (*n* = 1 [34], bimekizumab (*n* = 1) [30], CJM112 (*n* = 1) [35], etanercept (*n* = 1) [36], guselkumab (*n* = 1) [37], IFX-1 (*n* = 1) [38], INCB054707 (*n* = 1) [39], infliximab (*n* = 1) [22], and MABp1 (*n* = 1) [40].

A total of 8 different types of clinical disease activity assessment tools and 14 different types of patient-reported outcome measures were used by the included RCTs. Clinical disease activity assessment tools included HiSCR (*n* of RCTs = 12) [21,28,30,31,32,33,34,37,38,39,40], MSS (*n* = 6) [21,28,38,39,40], IHS4 (*n* = 3) [30,38,39], HS-PGA (*n* = 2) [21,35], PGA (*n* = 2) [22,36], SS (*n* = 2) [29,32], Hurley score (*n* = 1) [29], and hidradenitis suppurativa severity index (*n* = 1) [22]. Patient-reported outcomes included DLQI (*n* of RCTs = 12) [21,22,28,29,30,31,32,33,36,37,38], visual analogue scale (VAS) (*n* = 5) [21,22,29,32,40], patients’ global assessment (PtGA) (*n* = 5) [28,30,31,38], hospital anxiety and depression scale (*n* = 2) [28,37], numeric rating scale (NRS) (*n* = 2) [33,39], treatment satisfaction questionnaire for medication (*n* = 2) [28], work productivity and activity impairment questionnaire: specific health problem (*n* = 2) [28], EQ-5D (*n* = 1) [28], hidradenitis suppurativa impact assessment (*n* = 1) [31], hidradenitis suppurativa-investigator’s global assessment (*n* = 1) [37], hidradenitis suppurativa quality of life score (*n* = 1) [39], hidradenitis suppurativa symptom assessment (*n* = 1) [6], hidradenitis suppurativa symptom diary (*n* = 1) [37], and short-form 36 health status survey (*n* = 1) [37]. The characteristics and results of treatment response are shown in Appendix A

### 3.1. Risk of Bias of Included Trials

As illustrated in Figure 2, most of the included RCTs were rated with low risk of bias in all domains. Five [34,35,36,38,40] and three [34,35,36] RCTs were rated with unclear risk of selection and detection biases, respectively, because of lacking details of randomization process and blinding of outcome assessment. Two RCTs were rated with high risk of attrition bias because the lost to follow-up rate exceeded 20% [22,39], and another two RCTs were rated with unclear risk because of lacking relevant information [34,35].

### 3.2. Hidradenitis Suppurativa Clinical Response at 12 to 16 Weeks

Eleven RCTs on eight treatments (adalimumab, anakinra, apremilast, avacopan, bimekizumab, guselkumab, IFX-1, and MABp1) provided efficacy data on HiSCR at weeks 12 to 16 [21,28,30,31,32,33,34,37,38,40]. As shown in Figure 3, the meta-analysis found only adalimumab (RR 1.77, 95% CI 1.44–2.17, I^2^ = 26%, five RCTs) and bimekizumab (RR 2.25, 95% CI 1.03–4.92, one RCT) were significantly superior to placebo in achieving HiSCR response, with no significant difference noted between adalimumab and bimekizumab (*p* = 0.56).

### 3.3. Serious Adverse Events

All the included RCTs reported safety data except that the Adams 2010 trial did not report the safety data of etanercept [21,22,28,29,30,31,32,33,34,35,36,37,38,39,40]. As shown in Figure 4, the meta-analysis found no differences among any of the active treatments and placebo in the risk for SAEs. The Glatt 2021 trial did not specify SAEs but found no differences in treatment-emerged adverse events between bimekizumab and placebo [30].

### 3.4. Dermatologic Life Quality Index

Six RCTs [21,28,29,30,31] assessed the mean improvement of DLQI score in HS patients treated by adalimumab, with five showing significantly better improvement of DLQI score compared with placebo [21,28,30,31]. Bimekizumab, guselkumab, and infliximab also showed similar results, while anakinra, apremilast, etanercept, and IFX-1 did not significantly differ from placebo [22,30,32,33,36,37,38]. As illustrated in Figure 5, the meta-analysis found adalimumab (RR 3.97; 95% CI 1.70–9.28, I^2^ = 0%, three RCTs) and bimekizumab (RR 15.23; 95% CI 0.95–242.79, one RCT) were both superior to placebo in achieving DLQI 0/1 but only adalimumab reached significant difference.

### 3.5. Sartorius Score and Modified Sartorius Score

The Miller 2011 trial found no significant difference between the adalimumab and placebo groups in the change of SS at week 12 [29]. The Kimball 2012 trial found no significant difference in improvement of MSS at week 16 between adalimumab and placebo [21]. The opposite results were noted between the PIONEER I and II trials in which significant difference between adalimumab and placebo at week 12 was only noted in the PIONEER II trial [28]. Anakinra, IFX-1, INCB054707, and MABp1 did not significantly differ from placebo in the change of either SS or MSS [32,38,39,40].

### 3.6. Physician Global Assessment and Hidradenitis Suppurativa Physician Global Assessment

Among the included RCTs, two versions of PGA (Adams 2010 and Grant 2010) and HS-PGA (Kimball 2012 and 2020) were used to evaluate the efficacy [21,22,28,36]. Adalimumab, CJM112, and infliximab showed significantly superior treatment response than placebo, whereas etanercept did not [21,22,28,36].

### 3.7. Pain

VAS, NRS, and PtGA were used to evaluate pain among the included RCTs. The effects of adalimumab varied among the included RCTs. The Miller 2011 trial reported no statistical difference in change of pain VAS score, but the Bechara 2021 trial and the Kimball 2012/2016 trials found significant improvement in pain measured by VAS and PtGA [21,28,29,31]. The Glatt 2021 trial found no significant differences among the adalimumab, bimekizumab, and placebo groups in achieving ≥ 30% decrease in the PtGA [30]. Apremilast, INCB054707, and infliximab showed significant improvement in pain compared with placebo, whereas anakinra, IFX-1, and MABp1 did not [22,32,33,38,39,40].

## 4. Discussion

Among all biologics and small-molecule inhibitors developed for treating moderate-to-severe HS, we found only adalimumab and bimekizumab consistently effective in improving both the disease severity and life quality with no increase in SAEs.

Various scoring systems have been developed to measure disease severity and treatment response of HS, including the Hurley staging system, MSS, HiSCR, HS-PGA, and DLQI. Except for the HiSCR with good-quality validation data, the other outcome measures lack validation and correlation analysis [24,41]. Most of the included RCTs used HiSCR as clinical disease activity assessment tools, followed by MSS. However, most RCTs reported *p* value only and a meta-analysis could only be performed on HiSCR. Despite the lack of sufficient data for further meta-analysis, the efficacy measured by various instruments were consistent across the included RCTs. However, though improvement in pain was found in some RCTs, there was no consistent improvement in other outcome measures [30,33,39]. Only four RCTs (Janssen Research & Development LLC 2021, Kimball 2012 trial, PIONEER I, and Grant) showed inconsistent outcomes among the tools other than the pain scale they used [21,22,28,37]. In the Janssen Research & Development LLC 2021 trial, the guselkumab 200 mg group reached significant improvement in DLQI but not in achieving HiSCR. DLQI is a patient-reported outcome measure instrument, which is not specific to HS. Additionally, subjective evaluation was inconsistent with objective evaluation. These may lead to the different results between HiSCR and DLQI [37]. In the Kimball 2012 trial, the weekly adalimumab group showed significant improvement in HS-PGA, HiSCR, and DLQI, but not in the MSS [21]. In the PIONEER I trial, the adalimumab group showed significant improvement in HiSCR and DLQI but not in the MSS; by contrast, the adalimumab group achieved significant improvement in all the three outcomes in the PIONEER II trial [28]. The MSS evaluates the number of fistula, which is not assessed in the HiSCR and not expected to change dramatically with adalimumab therapy. The participants in PIONEER I showed higher disease burden, including higher draining fistula counts and higher mean MSS at baseline when compared with participants in PIONEER II. These differences may lead to the different results in Kimball 2012 trial and between the PIONEER I and II trials. In the Grant 2010 trial, the infliximab group showed significant improvement in DLQI, PGA, and VAS but not in HSSI [22]. However, the HSSI has not been validated, and this could explain the inconsistent results between HSSI and other assessment tools [42]. The Miller 2011 trial used a lower dose of adalimumab than the approved regimen; therefore, the results differed from other included adalimumab trials using the approved regimen [21,28,29,30,31].

Follicular occlusion followed by follicular rupture and foreign-body type immune response initiates the development of HS. The cytokines and immune pathway involved in HS have not been fully elucidated. Although there is no consensus about which cytokines drive the inflammation in HS, TNF, interleukin (IL)-1β, IL-10, and IL-17 are key cytokines known to be involved in HS. Elevated levels of TNF, IL-1β, IL-10, IL-17, and particularly IL-17A have been detected in HS lesional skin [43,44,45,46]. A recent study using gene set variation analysis and single-cell RNA sequencing suggested that the immunopathogenesis of HS involves the IL-1 pathway and type 1 T cell responses [47]. Our analysis included biologics and small-molecule inhibitors targeting TNF (adalimumab, etanercept, and infliximab), IL-1 (anakinra and MABp1), IL-17 (bimekizumab and CJM112), PDE4 (apremilast), complement 5a (C5a), and C5a receptor (avacopan and IFX-1), and Janus kinase inhibitor (INCB054707).

A network meta-analysis on efficacy of nonsurgical monotherapy for HS, which included trials before August 19, 2020, was recently published [48]. Besides biologics and small-molecule inhibitors, the network meta-analysis also included clindamycin, tetracycline, and botulinum toxin type B (BTX-B) in comparison. However, mild HS patients were enrolled in some of the included RCTs [49,50]. The heterogeneous severity of participants and endpoints across the included trials led to the incomparability between the interventions and was unlikely to satisfy the transitivity assumption between the networks. For quality of life, the network meta-analysis showed a higher SUCRA of BTX-B than adalimumab. However, BTX-B was compared with the lower dose regimen of adalimumab instead of the approved regimen, which led to the result being questioned because the approved regimen has shown better efficacy than the lower dose regimen in previous RCTs [21]. Compared with the network meta-analysis, our study only included participants with moderate-to-severe HS because biologics and small-molecule inhibitors are indicated for these patients when conventional therapy fails. We excluded unapproved low-dose regimen of adalimumab from our meta-analysis. We also included the latest RCTs on avacopan and IFX-1 in our meta-analysis. We found only adalimumab and bimekizumab effective in treating HS when using HiSCR as assessment tool. CJM112, INCB054707, and infliximab were not included in the meta-analysis. CJM112 was reported significantly more effective than placebo when assessed treatment response by HS-PGA [35]; however, only abstract was available and more data are needed to validate the results. Both INCB054707 and infliximab also failed to reach significant difference compared with placebo in the trials [22,39]. Our study also found no significant increase in SAEs associated with biologics and small-molecule inhibitors. These safety data were consistent with other reported studies when used for different indications [51,52,53,54,55,56,57,58,59,60].

Currently, a large number of biologics and small molecule inhibitors are under development for HS, including those targeting IL-1 receptor-associated kinase 4 (PF-0665083), IL-17 (secukinumab), IL-17 receptor A (brodalumab), granulocyte colony-stimulating factor (CSL324), IL-23 p19 (risankizumab and guselkumab), tyrosine kinase (PF-06826647), TYK2/JAK1 (brepocitinib), Janus kinase (upadacitinib), CD40 (iscalimab), CXCR receptor (LY3041658), and leukotriene A_4_ hydrolase (LSY006) [11,61]. The safety and clinical response at 12 to 24 weeks of brodalumab in moderate-to-severe HS had been examined in a recent open-label cohort study [62]. Brodalumab was administered with a dose of 210 mg at weeks 0, 1, and 2 and every 2 weeks thereafter in 10 patients. All patients achieved HiSCR at week 2 and had a 75% reduction in abscess and nodule (AN) count at week 24. Eight patients (80%) achieved IHS4 category change at week 12. Other outcomes, including VAS of pain, itch, and global disease assessment, SS, DLQI, and patient health questionnaire-9, also had significant decrease from baseline at week 12. No grade 2/3 adverse events associated with the use of brodalumab were reported during the study. Efficacy of secukinumab in subjects with moderate-to-severe HS had also been examined in an open-label trial (SUNRISE) [63]. Secukinumab was administered at 2 dose levels, a loading dose of five 300 mg injections weekly, followed by 300 mg every 4 weeks (9 patients) or 300 mg every 2 weeks (11 patients). A total of 65% (13 patients) and 70% (14 patients) achieved HiSCR at weeks 12 and 14 individually. Significant reduction of DLQI score from baseline was noted at week 12 rather than week 24. No SAE occurred during the trial and only results of entire study population were available currently. Further head-to-head trials are needed to compare the efficacy of systemic drugs.

Our analysis had several limitations. First, we could only perform meta-analysis on HiSCR and DLQI 0/1 because of lacking data on other efficacy outcomes. Second, most biologics and small-molecule inhibitors were only tested in a single RCT. The small sample size may lead to failure of bimekizumab in reaching statistically significant difference in DLQI 0/1. Third, nearly all included RCTs were phase 2, except adalimumab, which was tested in large-scale phase 3 and 4 trials.

## 5. Conclusions

To our knowledge, this study is the latest meta-analysis to compare the efficacy and safety of biologics and small-molecule inhibitors in treating HS. Adalimumab and bimekizumab are the only two effective biologics in achieving HiSCR with acceptable safety profile, whereas adalimumab is the only biologic effective in achieving DLQI 0/1.

## Figures and Tables

**Figure 1 biomedicines-10-01303-f001:**
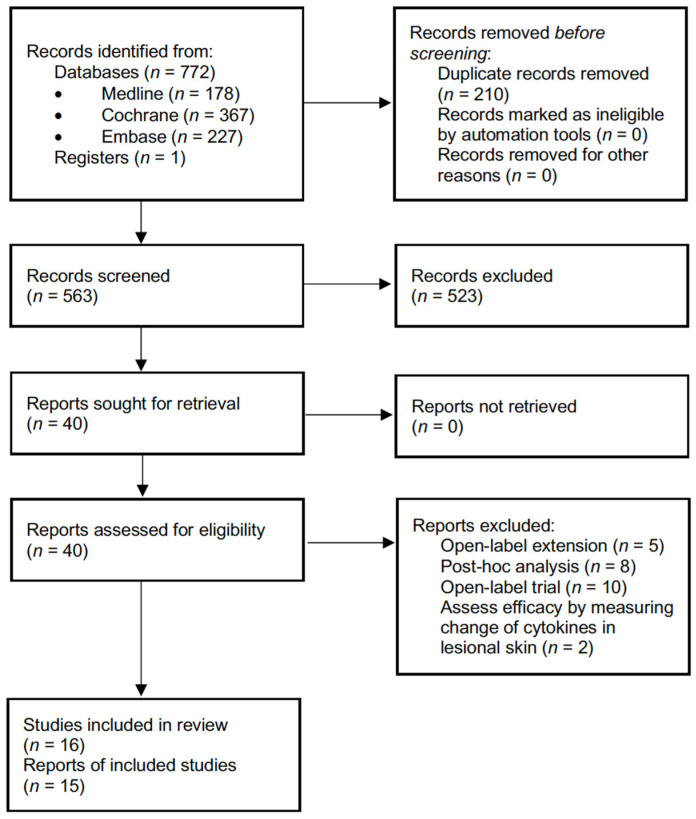
Preferred Reporting Items for Systematic review and Meta-Analyses (PRISMA) of study selection and inclusion.

**Figure 2 biomedicines-10-01303-f002:**
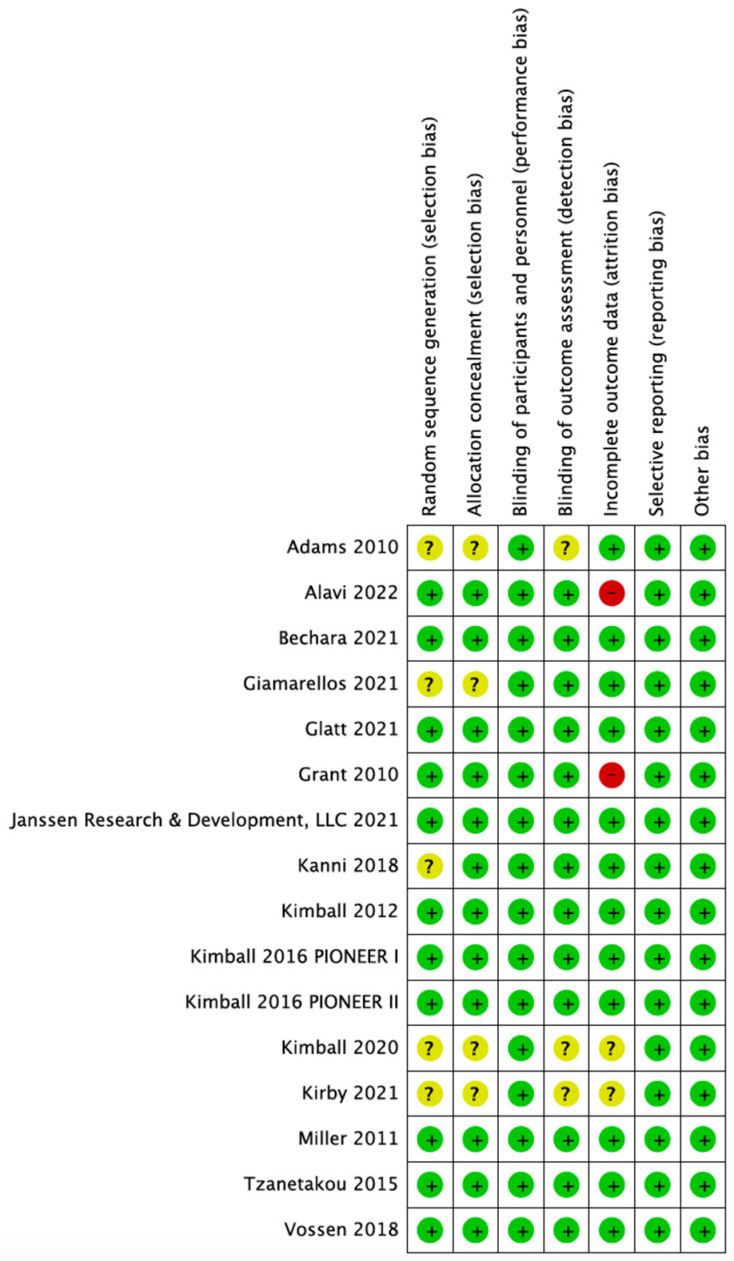
Risk of bias of included studies. A green dot denotes low risk of bias, yellow for unclear risk of bias, and red for high risk of bias.

**Figure 3 biomedicines-10-01303-f003:**
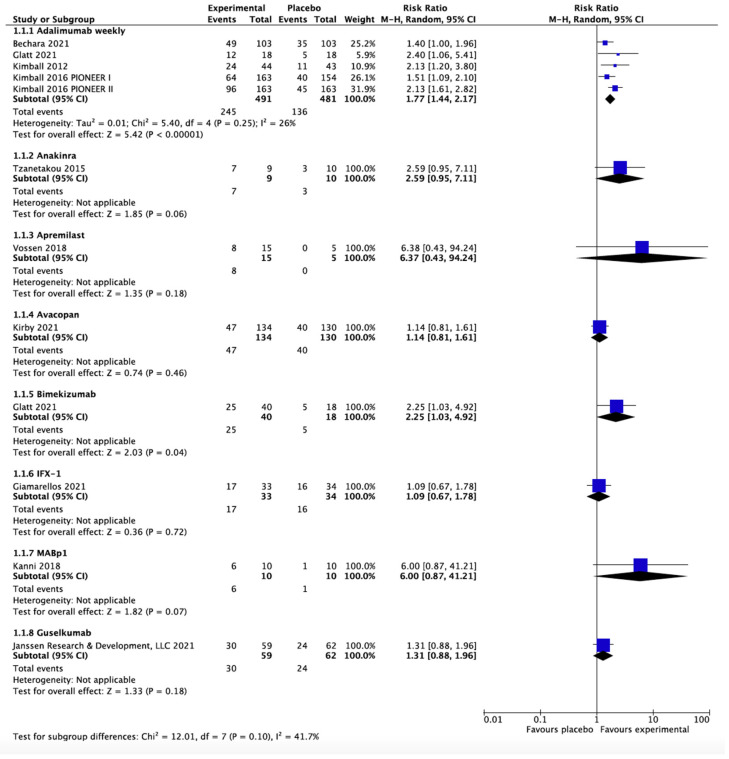
Hidradenitis suppurativa clinical response at weeks 12 to 16. The meta-analysis illustrated adalimumab (risk ratio 1.77, 95% confidence interval 1.50–2.09, I^2^ = 26%, 5 randomized controlled trials) and bimekizumab (risk ratio 2.25, 95% confidence interval 1.03–4.92, 1 randomized controlled trial) are the only two effective biologics in treating moderate-to-severe hidradenitis suppurativa.

**Figure 4 biomedicines-10-01303-f004:**
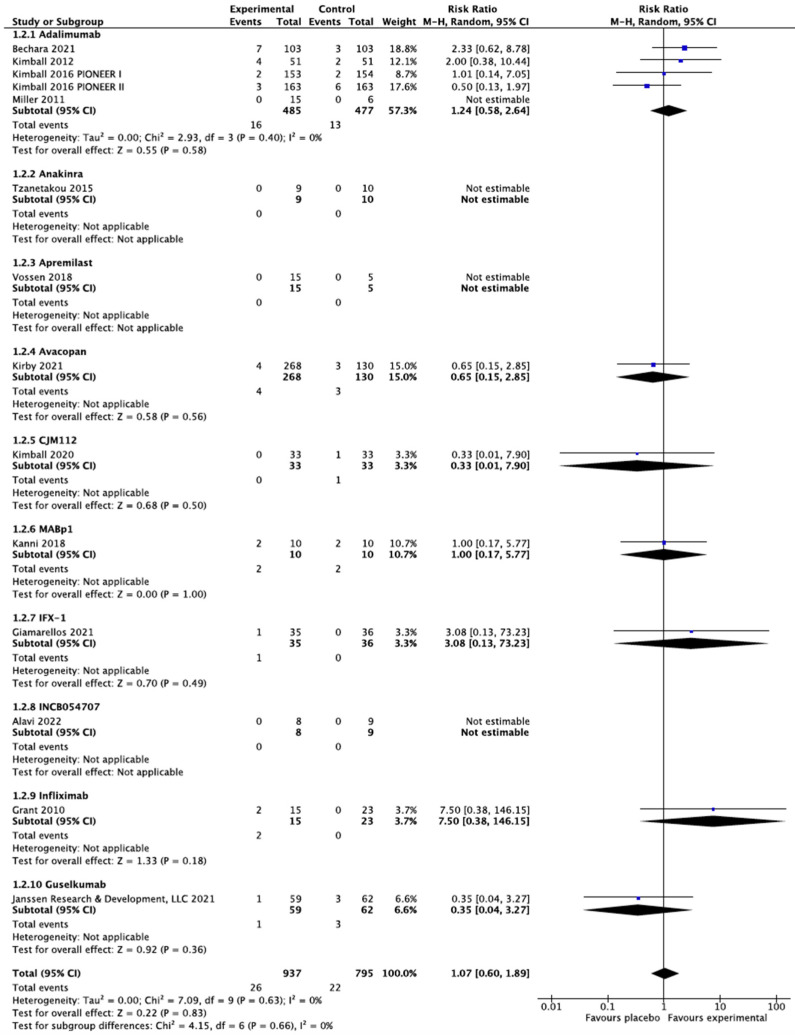
Serious adverse events among included biologics and small-molecule inhibitors. The meta-analysis illustrated no increase in serious adverse events compared with placebo among our included active treatments.

**Figure 5 biomedicines-10-01303-f005:**
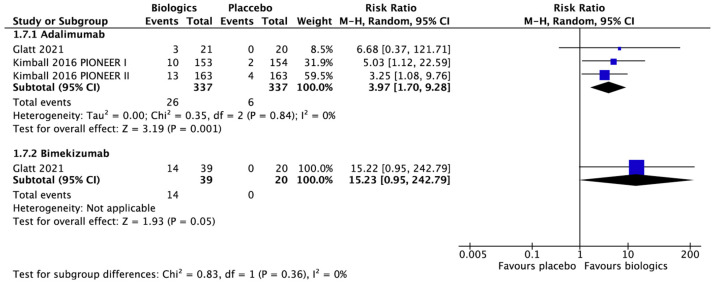
Dermatology life quality index 0/1 at weeks 12 to 16. The meta-analysis illustrated only adalimumab (risk ratio 3.97, 95% confidence interval 1.70–9.28, I^2^ = 0%, 3 randomized controlled trials) reached significant difference in achieving dermatology life quality index 0/1 at weeks 12 to 16.

## Data Availability

All data were collected from published articles available in the public domain or from clinicaltrials.gov (accessed on 19 May 2022).

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
