# Peer review of "Biologics and Small Molecule Inhibitors for Treating Hidradenitis Suppurativa: A Systematic Review and Meta-Analysis"

_biomedicines, 2022, doi:10.3390/biomedicines10061303_

Round 1

Reviewer 1 Report

A very interesting review and metanalysis about current available treatments for hidradenitis suppurativa, showing that adalimumab and bimekizumab are the two most promising drugs to manage this condition. The limitation of the study included the fact that only meta-analysis on HiSCR and DLQI 0/1 were included,  because of lacking data on other efficacy outcomes. Second, most  biologics and small molecule inhibitors were only tested in a single trial. I think the paper will be however, eligible to be published after some minor revisions:

line 51 you should add: also colchicine has been proposed in the management of HS and cite : doi: 10.3390/pharmaceutics14020294.

line 41: this sentence needs some references, such as: doi: 10.3390/ijms21228436.

Good Luck!!

Author Response

Dear Reviewer,

Many thanks for offering us an opportunity to revise our paper. Your comments are very useful and we have made corresponding changes. Below are our point-by-point reply to your comments.

Best wishes,

Ching-Chi Chi, MD, MMS, DPhil

  1. line 51 you should add: also colchicine has been proposed in the management of HS and cite: doi: 10.3390/pharmaceutics14020294.

Author’s reply: Many thanks for Reviewer's suggestion. We have added the sentence and citation (#19).

  1. line 41: this sentence needs some references, such as: doi:10.3390/ijms21228436.

Author’s reply: Many thanks for Reviewer's suggestion. We have added the reference (#10).

Reviewer 2 Report

This is a very well designed systematic review. Methods used are adequate,  results are interesting, and the manner in which they are reported is clear and exhaustive, the limits of the study are adequately pointed out.

Author Response

Many thanks for your compliments.